# Efficient Meshy Neural Fields for Animatable Human Avatars

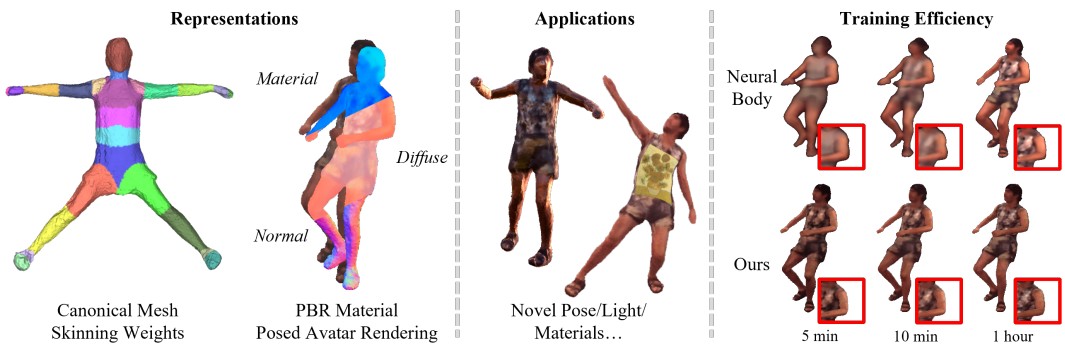

Figure 1: **EMA** efficiently and jointly learns canonical shapes, materials, and motions via differentiable inverse rendering in an end-to-end manner. The method does not require any predefined templates or riggings. The derived avatars are animatable and can be directly applied to the graphics rendering pipeline without any post-processing. **All figures are best viewed in color**.

## Abstract

Efficiently digitizing animatable human avatars from videos is a challenging and active research topic. Recent volume rendering-based neural representations open a new way for human digitization with their good usability and photo-realistic reconstruction quality. However, they are inefficient for long optimization times and slow inference speed; their implicit nature results in entangled geometry, materials, and dynamics of humans, which are hard to edit afterward. Such drawbacks prevent their direct applicability to downstream tasks, especially the prominent rasterization-based graphics pipeline. We present **EMA**, a method that **E**fficiently learns **M**eshy neural fields to reconstruct animatable human **A**vatars. It jointly optimizes explicit triangular canonical mesh, spatial-varying material, and motion dynamics, via inverse rendering in an end-to-end fashion. Each above component is derived from separate neural fields, relaxing the requirement of a template, or rigging. The mesh representation is highly compatible with the efficient rasterization-based renderer, thus our method only takes about an hour of training and can render in real-time. Moreover, only minutes of optimization are enough for plausible reconstruction results. The textured meshes enable direct downstream applications. Extensive experiments illustrate the very competitive performance and significant speed boost against previous methods. We also showcase applications including novel pose synthesis, material editing, and relighting.

## 1 Introduction

Recent years have witnessed the rise of human digitization (Habermann et al., 2020; Alexander et al., 2010; Peng et al., 2021b; Alldieck et al., 2018a; Raj et al., 2021). This technology greatly impacts the entertainment, education, design, and engineering industries. There is a well-developed industry solution for this task, which leverages **dense observations**. Photo-realistic reconstruction of humans can be achieved either with full-body laser scans (Saito et al., 2021), dense synchronized multi-view cameras (Xiang et al., 2021b; 2022; 2021a), or light stages (Alexander et al., 2010). However, these

settings are expensive, tedious to deploy, and consist of a complex processing pipeline, preventing the technology's democratization.

Another solution is to formulate the problem as inverse rendering and learn digital humans directly from **sparse observations**. Traditional approaches directly optimize explicit mesh representation (Loper et al., 2015; Fang et al., 2017; Pavlakos et al., 2019) which suffers from the problems of smooth geometry and coarse textures (Prokudin et al., 2021; Alldieck et al., 2018b). Besides, they require professional artists to design human templates, rigging, and unwrapped UV coordinates. Recently, with the help of volumetric-based implicit representations (Mildenhall et al., 2022; Park et al., 2019; Mescheder et al., 2019) and neural rendering (Laine et al., 2020; Liu et al., 2019; Thies et al., 2019), one can easily digitize a photo-realistic human avatar from sparse multi-view, or even single-view video(s) (Jiang et al., 2022; Weng et al., 2022). Particularly, volumetric-based implicit representations (Mildenhall et al., 2022; Peng et al., 2021b) can reconstruct scenes or objects with higher fidelity against previous neural renderer (Thies et al., 2019; Prokudin et al., 2021), and is more user-friendly as it does not need any human templates, pre-set rigging, or UV coordinates. The captured footage and its corresponding skeleton tracking are enough for training. However, better reconstructions and better usability are at the expense of the following factors. 1) **Inefficiency in terms of rendering**: They require longer optimization times (typically tens of hours or days) and inference slowly. Volume rendering (Mildenhall et al., 2022; Lombardi et al., 2019) formulates images by querying the densities and colors of millions of spatial coordinates. In the training stage, only a tiny fraction of points are sampled due to memory constraints, which leads to slow convergence speed. 2) **Inefficiency in terms of skinning**: Previous methods leverage existing the skinning template (Loper et al., 2015) which is not trainable and leads to sub-optimal results. To learn the subject-specific skinning, Chen et al. (2021b) proposed a root finding-based method to query the canonical points, which results in better quality but drastically increases the training time. 3) **Graphics incompatibility**: Volume rendering is incompatible with the current popular graphic pipeline, which renders triangular/quadrilateral meshes efficiently with the rasterization technique. Many downstream applications require mesh rasterization in their workflow (*e.g.*, editing (Foundation), simulation (Bender et al., 2014), real-time rendering (Möller et al., 2008), ray-tracing (Wald et al., 2019)). Although there are approaches (Lorensen & Cline, 1987; Labelle & Shewchuk, 2007) can convert volumetric fields into meshes, the gaps from discrete sampling degrade the output quality in terms of both geometry and textures.

To address these issues, we present **EMA**, a method based on **E**fficient **M**eshy neural fields to reconstruct animatable human **A**vatars. Our method enjoys **flexibility from implicit representations** and **efficiency from explicit meshes**, yet still maintains photo-realistic reconstruction quality. Given video sequences and the corresponding pose tracking, our method digitizes humans in terms of canonical triangular meshes, physically-based rendering (PBR) materials, and skinning weights *w.r.t.* skeletons. We jointly learn the above components via inverse rendering (Laine et al., 2020; Chen et al., 2021a; 2019) in an end-to-end manner. Each of them is encoded by a separate neural field, which relaxes the requirements of a preset human template, rigging, or UV coordinates. Specifically, we predict a canonical mesh out of a signed distance field (SDF) by differentiable marching tetrahedra (Shen et al., 2021; Gao et al., 2022; 2020; Munkberg et al., 2022), then we extend the marching tetrahedra (Shen et al., 2021) for spatial-varying materials by utilizing a neural field to predict PBR materials *on the mesh surfaces* after rasterization (Munkberg et al., 2022; Hasselgren et al., 2022; Laine et al., 2020). To make the canonical mesh animatable, we use another neural field to model the **forward linear blend skinning** for the canonical meshes. Thus, given a posed skeleton, the canonical meshes are transformed into the corresponding poses. Finally, we shade the mesh with a rasterization-based differentiable renderer (Laine et al., 2020) and train our models with a photo-metric loss. After training, we export the mesh with materials and discard the neural fields.

There are several merits of our method design. 1) **Efficiency in terms of rendering**: Powered by efficient mesh rendering, our method can render in real-time. Besides, the training speed is boosted as well, since we compute loss holistically on the whole image and the gradients only flow on the mesh surface. In contrast, volume rendering takes limited pixels for loss computation and back-propagates the gradients in the whole space. Our method only needs about an hour of training yet minutes of optimization are enough for plausible reconstruction quality. Our shape, materials, and motion modules are split naturally by design, which facilitates editing. To further improve reconstruction quality, we additionally optimize image-based environment lights and non-

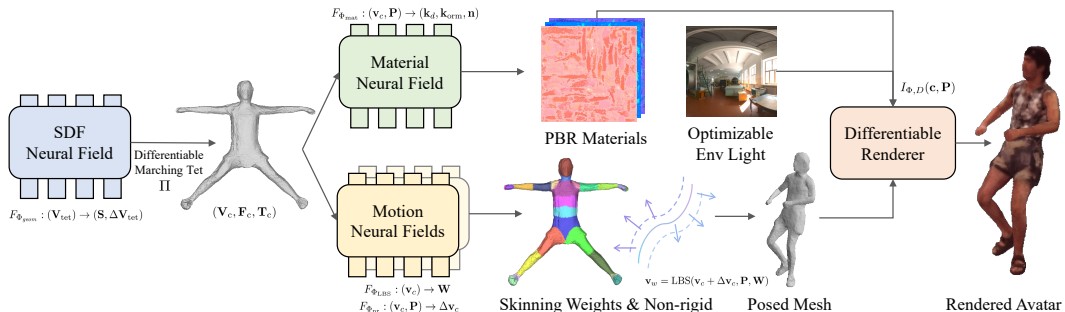

Figure 2: **The pipeline of EMA**. EMA jointly optimizes canonical shapes, materials, lights, and motions via efficient differentiable inverse rendering. The canonical shapes are attained firstly through the differentiable marching tetrahedra (Gao et al., 2020; Shen et al., 2021; Munkberg et al., 2022), which converts SDF fields into meshes. Next, it queries PBR materials, including diffuse colors, roughness, and specularity on the mesh surface. Meanwhile, the skinning weights and per-vertices offsets are predicted on the surface as well, which are then applied to the canonical meshes with the guide of input skeletons. Finally, a rasterization-based differentiable renderer takes in the posed meshes, materials, and environment lights, and renders the final avatars efficiently.

rigid motions. 2) **Efficiency in terms of skinning**: Compared with (Chen et al., 2021b), the root finding-based method to query the canonical points to learn forward skinning, Our method leverages differentiable rasterization to learn both mesh properties and the forward skinning jointly in an end-to-end optimizable way, which is orders of magnitude faster. Besides, Canonical meshes with forward skinning modeling handle the out-of-distribution poses better. 3) **Graphics compatibility**: Our derived mesh representation is compatible with the prominent graphic pipeline, which leads to instant downstream applications (*e.g.*, the shape and materials can be edited directly in the design software (Foundation)).

We conduct extensive experiments on standards benchmarks H36M (Ionescu et al., 2014) and ZJU-MoCap (Peng et al., 2021b). Our method achieves very competitive performance for novel view synthesis, generalizes better for novel poses, and significantly improves both training time and inference speed against previous arts. Our research-oriented code reaches real-time inference speed (100+ FPS for rendering $512 \times 512$ images). We in addition showcase applications including novel pose synthesis, material editing, and relighting.

## 2 RELATED WORKS

**Explicit Representations for Human Modeling:** It is intuitive to model the surfaces of humans with mesh. However, humans are highly varied in both shape and appearance and have complex pose distributions, which all contribute to a high-dimensional modeling space. To start with, researchers tried to model humans with statistical models and an assumption of nearly no clothes. One of the prevalent methods is parametric models (Anguelov et al., 2005; Loper et al., 2015; Pavlakos et al., 2019; Romero et al., 2017; Su et al., 2021). The models are learned with principal component analysis from enormous scans of humans with limited clothes (Loper et al., 2015). However, fitting humans from scans is inapplicable in real-world applications. Thus, Kanazawa et al. (2018); Bogo et al. (2016); Kocabas et al. (2021); Zhang et al. (2021a; 2022); Kocabas et al. (2020); Sun et al. (2021) proposed to estimate the human surface from images or videos. To model the clothed human, Prokudin et al. (2021); Alldieck et al. (2022; 2018b) deform the template human vertices in canonical T-pose. Nevertheless, these methods are prone to capturing coarse geometry due to the limited geometry budget and the weak deformation model. Besides, the textures are modeled with sphere harmonics which are far from photo-realistic. Our method takes the mesh as our core representation to enable efficient training and rendering, then achieves the topological change of shape and photo-realistic texture via neural fields.

**Implicit Representations for Human Modeling:** Implicit representations (Park et al., 2019; Mescheder et al., 2019; Mildenhall et al., 2022) model the objects in continuous functions, where

those explicit entities cannot be attained directly. The prevalent options are Signed Distance Function (Park et al., 2019), Occupancy Field (Mescheder et al., 2019) and Radiance Field (Mildenhall et al., 2022), which can be easily parameterized by neural networks. Given full-body scans as 3D supervision, Saito et al. (2019; 2020); He et al. (2021); Huang et al. (2020); Alldieck et al. (2022) learned the SDFs or occupancy fields directly from images. After training, they directly predict photo-realistic human avatars in inference time. Peng et al. (2021b); Su et al. (2021); Liu et al. (2021); Peng et al. (2021a); Li et al. (2022); Jiang et al. (2022); Chen et al. (2022); Wang et al. (2022); Zhang & Chen (2022); Noguchi et al. (2021); Zheng et al. (2022); Jiang et al. (2023) leveraged the radiance field to reconstruct photo-realistic human avatars from multi-view images or single-view videos without any 3D supervision. Although implicit representations improve reconstruction quality against explicit ones, they still have drawbacks, *e.g.*, large computation burden or poor geometry. Besides, volume rendering is presently under-optimized with graphics hardware, thus the outputs are inapplicable in downstream applications **without further post-processing**, which is either extremely slow or loses certain features like pose-dependent deformation. Our method absorbs the merits of implicit representations by using neural networks to predict canonical shape, pose-dependent deformation, and photo-realistic textures, leveraging (Shen et al., 2021) to convert SDFs to explicit meshes whose rendering is highly optimized within the graphics pipeline.

**Hybrid Representations for Human Modeling:** There are two tracks of literature modeling humans with explicit geometry representations and implicit texture representations. One track of literature (Khakhulin et al., 2022; Zhao et al., 2022) leveraged neural rendering techniques (Thies et al., 2019). Meshes (Prokudin et al., 2021; Zhao et al., 2022; Alldieck et al., 2018b;a; Xiang et al., 2021a), point clouds (Wang et al., 2021; Uzolas et al., 2023), or mixture of volumetric primitives (Remelli et al., 2022) are commonly chosen explicit representations. Moreover, fine-grained geometry and textures are learned by neural networks. However, these methods are either only applicable for novel view synthesis (Wang et al., 2021) or restricted to self-rotation video captures (Alldieck et al., 2018b;a). Besides, the neural renderers have limited capabilities leading to problems such as stitching texture (Karras et al., 2021; 2020), or baked textures inside the renderer. In contrast, the human avatars learned by our method are compatible with graphics pipeline, indicating that they are **directly applicable in downstream tasks**, *e.g.*, re-posing, editing in design software. The other track of literature took neural networks to learn both geometry and textures with differentiable rendering (Laine et al., 2020; Chen et al., 2021a; 2019). It equips the traditional graphics pipeline with the ability of gradient backpropagation (Liu et al., 2019; Blanz & Vetter, 1999; Laine et al., 2020; Ravi et al., 2020; Lassner & Zollhöfer, 2021; Raj et al., 2021)., which makes scene properties (*e.g.,* assets, lights, cameras poses, *etc.*) optimizable through gradient descent *w.r.t* photo-metric loss. Thus, both geometry and textures are learned in a way that is compatible with existing graphics hardware. However, the geometry optimization process is non-convex and highly unstable (Grassal et al., 2022), thus it is hard to produce fine-grained geometry details. Besides, the fixed and limited topology of the mesh results in limited capability of shape modeling. We convert SDFs to meshes with differentiable marching tets (Munkberg et al., 2022; Shen et al., 2021), and model the motion dynamics with an additional neural field. Our method enjoys flexibility from implicit representations and efficiency brought by explicit meshes, yet still maintains photo-realistic reconstructions.

## 3 METHOD

We formulate the problem as inverse rendering and extend Munkberg et al. (2022) to model dynamic actors that are driven solely by skeletons. The canonical shapes, materials, lights, and actor motions are learned jointly in an end-to-end manner. The rendering happens with an efficient rasterization-based differentiable renderer (Laine et al., 2020).

**Optimization Task**: Let $\Phi$ denote all the trainable parameters in neural networks that encode: (1) SDF values and corresponding offsets defined on the tet vertices for canonical geometry; (2) spatial-varying and pose-dependent materials and environmental light probe for shading; (3) forward skinning weights and non-rigid offsets defined on the mesh vertices for motion modeling.

For a given camera pose $\mathbf{c}$ and a tracked skeleton pose $\mathbf{P}$, we render the image $I_{\Phi,D}(\mathbf{c}, \mathbf{P})$ with a differentiable renderer $D$, and compute loss with a loss function $L$, against the reference image

$I_{ref}(\mathbf{c}, \mathbf{P})$. The optimization goal is to minimize the empirical risk:

$$\underset{\phi}{\arg\min} \; \mathbb{E}_{\mathbf{c}, \mathbf{P}} \left[ L \big( I_{\Phi, D}(\mathbf{c}, \mathbf{P}), I_{ref}(\mathbf{c}, \mathbf{P}) \big) \right]. \tag{1}$$

The parameters $\Phi$ are optimized with Adam (Kingma & Ba, 2015) optimizer. Following (Munkberg et al., 2022), our loss function $L = L_{\text{img}} + L_{\text{mask}} + L_{\text{reg}}$ consists of three parts: an image loss $L_{img}$ using $\ell_1$ norm on tone mapped color, and mask loss $L_{\text{mask}}$ using squared $\ell_2$, and regularization losses $L_{\text{reg}}$ to improve the quality of canonical geometry, materials, lights, and motion.

**Difference between rasterization-based rendering and volume rendering**: At each optimization step, our method holistically learns both shape and materials from **the whole image**, while the volume rendering-based implicit counterparts only learn from **limited points per pixels**. Besides, the gradients in our method only flow through the iso-surface, which is drastically less than NeRF, whose gradients flow over the whole space. Thus, our method can converge even faster.

Powered by an efficient rasterization-based renderer, our method enjoys both faster convergence and real-time rendering speed. For **details about optimization and losses**, please refer to our **supplementary**.

### 3.1 CANONICAL GEOMETRY

Rasterization-based differentiable renderers take triangular meshes as input, which means the whole optimization process happens over the mesh representation. Previous works (Alldieck et al., 2018b;a) require a mesh template to assist optimization as either a shape initialization or regularization. The templates that have fixed topology and limited resolutions harm the geometry quality. Besides, to make the learned geometry generalize to novel poses, the underlying geometry representations should lie in a canonical space.

We utilize the differentiable marching tetrahedra (Shen et al., 2021; Gao et al., 2020) algorithm to model the humans in **canonical space**, which converts SDF fields into triangular meshes. This method enjoys the merit of being **template-free and topology-free** from **implicit** SDF representations, then produces **explicit** triangle meshes that are **efficient and directly applicable** to rasterization-based renderers.

Let $\mathbf{V}_{\text{tet}}, \mathbf{F}_{\text{tet}}, \mathbf{T}_{\text{tet}}$ and be the pre-defined vertices, faces, and UV coordinates of the tetrahedra grid. We parameterize both per-tet vertice SDF value $\mathbf{S}$ and vertices offsets $\Delta \mathbf{V}_{\text{tet}}$ with a coordiante-based nerual network:

$$F_{\Phi_{geom}} : (\mathbf{V}_{\text{tet}}) \rightarrow (\mathbf{S}, \Delta \mathbf{V}_{\text{tet}}), \tag{2}$$

the canonical mesh $\mathbf{M}_{\text{c}} = (\mathbf{V}_{\text{c}}, \mathbf{F}_{\text{c}}, \mathbf{T}_{\text{c}})$ (*i.e,* canonical mesh vertices, faces, and UV map coordinates) is derived by marching tetrahedra operator $\Pi$:

$$\Pi : (\mathbf{V}_{\text{tet}}, \mathbf{F}_{\text{tet}}, \mathbf{T}_{\text{tet}}, \mathbf{S}, \Delta \mathbf{V}_{\text{tet}}) \rightarrow (\mathbf{V}_{\text{c}}, \mathbf{F}_{\text{c}}, \mathbf{T}_{\text{c}}). \tag{3}$$

Specifically, the vertices of the canonical mesh are computed by $\mathbf{v}_c^{ij} = \frac{\mathbf{v}_{tet}'^{i} s_j - \mathbf{v}_{tet}'^{j} s_i}{s_j - s_i}$, where $\mathbf{v}_{tet}'^{i} = \mathbf{v}^i + \Delta \mathbf{v}^i$ and $\text{sign}(s_i) \neq \text{sign}(s_j)$. In other words, only the edges that cross the surface of canonical mesh participate the marching tetrahedra operator, which makes the mesh extraction both computation and memory-efficient.

Besides, compared with the volume rendering-based method whose gradient flows all over the space, the gradient flow of our method only passes through the iso-surface, which makes our model **converge even faster**. As well for rendering and loss computation overhead, our method can render the whole image for loss computing, which **further improves the training efficiency**. While for volume rendering counterparts, only a small fraction of pixels are involved.

To reduce the memory usage at the beginning of training, we initialize the SDF field to match the **coarse visual hull** of humans. After training, we can discard the SDF and deformation neural nets $F_{\Phi_{geom}}$ and store the derived meshes. That leads to zero computation overhead in inference time. For more **details about geometry modeling (*e.g.*, initialization**, please refer to our **supplementary**.

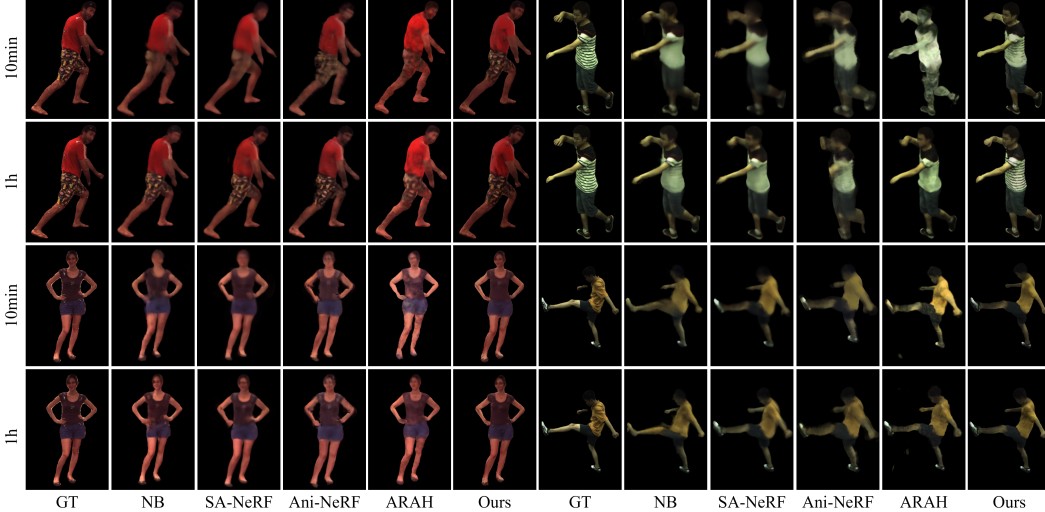

Figure 3: **Qualitative results of novel view synthesis on the H36M and ZJU-MoCap datasets**. (Peng et al., 2021b;a) generates blurry textures compared with our method. The mesh representations and forward skinning modeling help to improve rendering quality. Left: H36M dataset. Right: ZJU-MoCap dataset. **Zoom in for a better view**.

## 3.2 SHADING MODEL

**Materials**: we use a Physically-Based Rendering (PBR) material model (McAuley et al., 2012), which is directly applicable to the differentiable renderer. PBR material is a well-developed technique that is widely used in industries as gaming, movie, design, etc (McAuley et al., 2012). It consists of a diffuse term with an isotopic GGX lobe representing specularity. Concretely, it consists of three parts: 1) diffuse lobe $\mathbf{k}_d$ has four components, *i.e.* RGB color channels and an additional alpha channel; 2) specular lobe comprises a roughness value $r$ for GGX normal distribution function and a metalness factor $m$ which interpolates the sense of reality from plastic to pure metallic appearance. The specular highlight color is given by an empirical formula $\mathbf{k}_s = (1 - m) \cdot 0.04 + m \cdot \mathbf{k}_d$. We store the specular lobe into texture $\mathbf{k}_{\mathrm{orm}} = (o, r, m)$, where the channel $o$ is unused by convention. To compensate for the global illumination, we alternatively store the ambient occlusion value into $o$. 3) normal maps $\mathbf{n}$ represents the fine-grained geometry details. The diffues color $\mathbf{k}_d$, texture $\mathbf{k}_{\mathrm{orm}}$, and normal maps $\mathbf{n}$ are parametrized by an neural network:

$$F_{\Phi_{\mathrm{mat}}} : (\mathbf{v}_c, \mathbf{P}) \rightarrow (\mathbf{k}_d, \mathbf{k}_{\mathrm{orm}}, \mathbf{n}). \tag{4}$$

Following standard rasterization-based rendering, We query the material values given the vertices after rasterization and barycentric interpolation **over the canonical mesh**. The PBR material is further conditioned on poses to model the pose-dependent shading effect.

**Lights**: Our method learns a fixed environment light directly from the reference images (Munkberg et al., 2022). The lights are represented as a cube light. Given direction $\omega_o$, We follow the render equation (Kajiya, 1986) to compute the outgoing radiance $L(\omega_o)$:

$$L(\omega_o) = \int_{\Omega} L_i(\omega_i) f(\omega_i, \omega_o) (\omega_i \cdot \mathbf{n}) d\omega_i, \tag{5}$$

the outgoing radiance is the integral of the incident radiance $L_i(\omega_i)$ and the BRDF $f(\omega_i, \omega_o)$. We do not use spherical Gaussians (Chen et al., 2019) or spherical harmonics (Boss et al., 2021; Zhang et al., 2021b) to approximate the image-based lighting. Instead, we follow (Munkberg et al., 2022) using the split sum approximation that capable of modeling all-frequency image-based lighting:

$$L(\omega_o) \approx \int_{\Omega} f(\omega_i, \omega_o) (\omega_i \cdot \mathbf{n}) d\omega_i$$
$$\int_{\Omega} L_i(\omega_i) D(\omega_i, \omega_o) (\omega_i \cdot \mathbf{n}) d\omega_i. \tag{6}$$

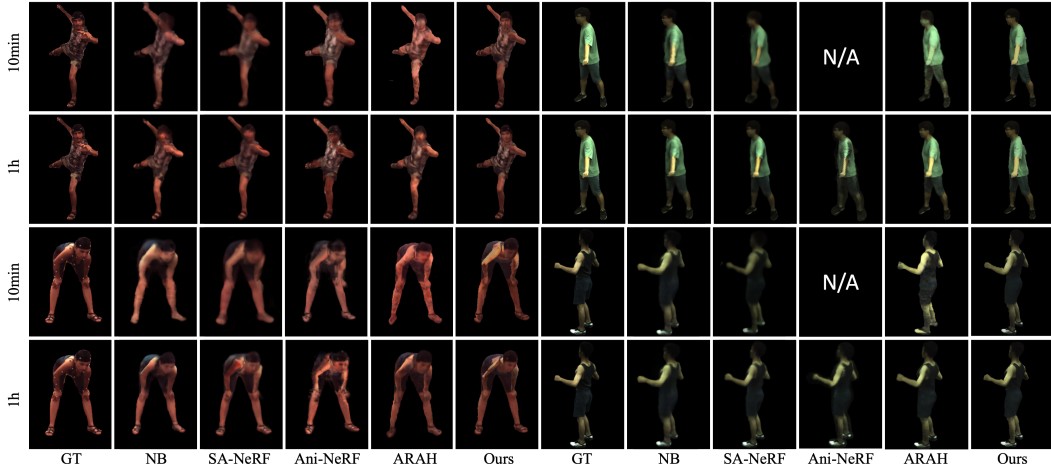

Figure 4: **Qualitative results of novel pose synthesis on the H36M and ZJU-MoCap datasets**. (Peng et al., 2021b;a) generates blurry textures compared with our method. The mesh representations and forward skinning modeling help to improve generalization. Left: H36M dataset. Right: ZJU-MoCap dataset. **Zoom in for a better view**.

The materials and lights are optimized jointly with geometry and motion modules in an end-to-end fashion. The decomposed design of geometry and shading, along with compatibility with the triangle renderer enables editing and content creation instantly after training.

For more **details about the shading and lighting model**, please refer to the **supplementary**.

### 3.3 MOTION MODEL

After we define the mesh-based avatars in canonical space with the PBR materials and the environmental lights, it is intuitive and natural to choose **forward linear skinning** (Rumman & Fratarcangeli, 2016) as our motion model. Given a skeleton with $B$ bones, the skeleton poses $\mathbf{P} = \{\mathbf{T}_1, \mathbf{T}_2, \ldots, \mathbf{T}_B\}$, where each $\mathbf{T}_i$ represents the transformation on bone $i$, and the blend skinning weights $\mathbf{W} = \{w_1, w_2, \ldots, w_B\}$, we deform each mesh vertice $\mathbf{v}_c$ in canonical space to the posed vertice $\mathbf{v}_w$ in world space by:

$$\mathbf{v}_w = \mathrm{LBS}(\mathbf{v}_c, \mathbf{P}, \mathbf{W}) = (\sum_{i=1}^{B} w_i \mathbf{T}_i)\mathbf{v}_c, \tag{7}$$

to compensate for non-rigid cloth dynamics, we add a layer of pose-dependent non-rigid offsets $\Delta\mathbf{v}_c$ for canonical meshes:

$$\mathbf{v}_w = \mathrm{LBS}(\mathbf{v}_c + \Delta\mathbf{v}_c, \mathbf{P}, \mathbf{W}), \tag{8}$$

where the blend skinning weights and the pose-dependent non-rigid offsets are, respectively, parameterized by neural networks whose inputs are **canonical mesh vertices**:

$$F_{\Phi_{\mathrm{LBS}}} : (\mathbf{v}_c) \rightarrow \mathbf{W}, \tag{9}$$

$$F_{\Phi_{\mathrm{nr}}} : (\mathbf{v}_c, \mathbf{P}) \rightarrow \Delta\mathbf{v}_c. \tag{10}$$

Modeling forward skinning is **efficient** for training as it only forward once in each optimization step, while the volume-based methods (Li et al., 2022; Wang et al., 2022; Chen et al., 2021b) solve the root-finding problem for canonical points in every iteration, which is computationally expensive and time-consuming.

We leverage the existing skinning weights from the parametric model SMPL (Loper et al., 2015) to initialize and regularize the skinning neural field as (Wang et al., 2022). The offset field is as well regularized to be zero as (Li et al., 2022). After training, we can export the skinning weight from neural networks which removes the extra computation burden for inference. For more **details about the motion model**, please refer to our **supplementary**.

Table 1: **Quantitative results**. On the marker-based H36M, our method achieves SOTA performance in all optimization durations. While on the markerless ZJU-MoCap, our method is comparable with previous arts. "T.F." means template-free; "Rep." means representation; "T.T" means the training time; ∗ denotes the evaluation on a subset of validation splits. Though there exists a lag of quantitative performance on ZJU-Mocap due to the dataset quality which **breaks our constant lighting assumption** (we discuss this in supp.), the **visual quality is better** as shown in **Figure 3, 4**.

| | T.F. | Rep. | T.T. | H36M | | | | ZJUMOCAP | | | |
| | | | | Training pose | | Novel pose | | Training pose | | Novel pose | |
| | | | | PSNR↑ | SSIM↑ | PSNR↑ | SSIM↑ | PSNR↑ | SSIM↑ | PSNR↑ | SSIM↑ |
|---|---|---|---|---|---|---|---|---|---|---|---|
| NB Peng et al. (2021b) | | NV | ∼10 h | 23.31 | 0.902 | 22.59 | 0.882 | 28.10 | 0.944 | 23.49 | 0.885 |
| SA-NeRF Xu et al. (2022) | | NV | ∼30 h | 24.28 | 0.909 | 23.25 | 0.892 | 28.27 | 0.945 | 24.42 | 0.902 |
| Ani-NeRF Peng et al. (2021a) | ✓ | NV | ∼10 h | 23.00 | 0.890 | 22.55 | 0.880 | 26.19 | 0.921 | 23.38 | 0.892 |
| ARAH Wang et al. (2022) | ✓ | NV | ∼48 h | 24.79 | 0.918 | 23.42 | 0.896 | 28.51 | 0.948 | 24.63 | 0.911 |
| Ours | ✓ | Hybr | ∼1 h | 24.72 | 0.916 | 23.64 | 0.899 | 26.57 | 0.901 | 24.38 | 0.875 |
| NB | | NV | | 20.58 | 0.879 | 20.27 | 0.867 | 26.87 | 0.922 | 23.67 | 0.885 |
| SA-NeRF | | NV | | 21.03 | 0.878 | 20.71 | 0.869 | 24.92 | 0.882 | 23.38 | 0.869 |
| Ani-NeRF | ✓ | NV | ∼1 h* | 22.54 | 0.872 | 21.79 | 0.856 | 21.23 | 0.659 | 20.65 | 0.652 |
| ARAH | ✓ | NV | | 24.25 | 0.904 | 23.61 | 0.892 | 26.33 | 0.924 | 24.67 | 0.911 |
| Ours | ✓ | Hybr | | 24.83 | 0.917 | 23.64 | 0.899 | 26.66 | 0.901 | 24.64 | 0.880 |
| NB | | NV | | 20.54 | 0.863 | 20.15 | 0.853 | 25.37 | 0.894 | 23.54 | 0.873 |
| SA-NeRF | | NV | | 20.81 | 0.848 | 20.49 | 0.841 | 24.48 | 0.878 | 23.75 | 0.872 |
| Ani-NeRF | ✓ | NV | ∼10 m* | 20.57 | 0.822 | 20.22 | 0.806 | 21.17 | 0.652 | 21.16 | 0.656 |
| ARAH | ✓ | NV | | 23.83 | 0.895 | 23.13 | 0.884 | 25.09 | 0.906 | 24.21 | 0.898 |
| Ours | ✓ | Hybr | | 24.27 | 0.909 | 23.37 | 0.897 | 25.51 | 0.888 | 24.42 | 0.878 |

## 4 EXPERIMENTS

### 4.1 DATASET AND METRICS

**H36M** consists of 4 multi-view cameras and uses **marker-based** motion capture to collect human poses. For the marker-based motion capture system, several markers are attached to the subject to indicate the motion of joints during the capture. Then, the 3D coordinate of the joints for each frame can be recovered via triangulation. The obtained 3D joints can form the tracked poses that are **considerably accurate**. Each video contains a single subject performing a complex action. We follow (Peng et al., 2021a) data protocol which includes subject S1, S5-S9, and S11. The videos are split into two parts: **"training poses" for novel view synthesis** and **"Unseen poses" for novel pose synthesis**. Among the video frames, 3 views are used for training, and **the rest views** are for evaluation. The novel view and novel pose metrics are computed on rest views. We use the same data preprocessing as (Peng et al., 2021a).

**ZJU-MoCap** records 9 subjects performing complex actions with 23 cameras. The human poses are obtained with a **markerless** motion capture system, whose joints are estimated by 2D keypoint estimation algorithm (Cao et al., 2021), which are **noisy** and may **lack temporal consistency**. The 3D coordinates of joints are produced by triangulation as well. Thus the pose tracking is rather **noisier** compared with H36M. Likewise, there are two sets of video frames, '**training poses" for novel view synthesis** and **"Unseen poses" for novel pose synthesis**. 4 evenly distributed camera views are chosen for training, and the rest 19 views are for evaluation. The evaluation metrics are computed on rest views. The same data protocol and processing approaches are adopted following (Peng et al., 2021b;a).

**Metrics**. We follow the typical protocol in (Peng et al., 2021b;a) using two metrics to measure image quality: peak signal-to-noise ratio (PSNR) and structural similarity index (SSIM).

### 4.2 EVALUATION AND COMPARISON

**Methods and Settings**. We compare our method with template-based methods (Peng et al., 2021b; Xu et al., 2022) and template-free methods (Peng et al., 2021a; Wang et al., 2022). Here we list the average metric values **given different training times** (*i.e.*, 10 minutes, 1 hour, and fully converged time) in Table 1 to illustrate our very competitive performance and significant speed boost. 1) Tempelate-based methods. Neural Body (NB) (Peng et al., 2021b) learns a set of latent codes anchored to a deformable template mesh to provide geometry guidance. Surface-Aligned NeRF (SA-NeRF) (Xu et al., 2022) proposes projecting a point onto a mesh surface to align surface points and signed height to the surface. 2) Template-free methods. Animatable NeRF (Ani-NeRF) (Peng et al.,

2021a) introduces neural blend weight fields to produce the deformation fields instead of explicit template control. ARAH (Wang et al., 2022) combines an articulated implicit surface representation with volume rendering and proposes a novel joint root-finding algorithm.

**Comparisons with state-of-the-arts (SOTAs)**. Table 1 illustrates the quantitative comparisons with previous arts. Notably, our method achieves very competitive performance within much less training time. The previous volume rendering-based counterparts spend tens of hours of optimization time, while our method only takes an hour of training (for previous SOTA method ARAH (Wang et al., 2022), it takes about 2 days of training). On the marker-based H36M dataset, our method reaches the SOTA performance in terms of novel view synthesis on training poses. It outperforms previous SOTA (ARAH (Wang et al., 2022)) for novel view synthesis on novel poses, which indicates that our method can generalize better on novel poses. The significant boost in training speeds lies in, on the one hand, the core mesh representation which can be rendered efficiently with the current graphic pipeline (Laine et al., 2020). On the other hand, the triangular renderer uses less memory. Thus we can compute losses over the whole image to learn the representations holistically. In contrast, previous NeRF-based methods are limited to much fewer sampled pixels in each optimization step, which converge slowly and are computationally expensive.

On the markerless ZJU-Mocap dataset, our method falls behind for training poses novel view synthesis and ranks 3rd place in terms of unseen poses novel view synthesis among the competitors. This is because ZJU-Mocap **breaks our assumption of constant lighting**. To enable editing of texture without involving lighting, we do not use view-direction as a condition as in (Mildenhall et al., 2022). We discuss this in detail **in the supplementary**. Despite a lag in quantitative comparison (Table 1), for comparison of fully converged models, our method is **still better** than most previous methods and is **on par** with the SOTA method ARAH (Figure 3 and 4). We also encourage our readers to view our **supplementary video** for further assessment. The misalignment of qualitative and quantitative results indicates the limitation of current evaluation metrics solely based on similarities.

We evaluate each method under **the same optimization duration** in Table 1. For the extremely low inference speed of our competitor, we only evaluate at most 10 frames in each subject, and for ZJU MoCap we only choose another 4 evenly spaced cameras as the evaluation views. For both 1 hour and 10 minutes optimization time, our method outperforms other methods for both training poses and unseen poses novel view synthesis on the marker-based H36M dataset. On the markerless ZJU-Mocap dataset, our method is comparable with previous SOTA in terms of PSNR and SSIM for both evaluation splits. Again, Figure 3 and 4 shows that our method can achieve **more favorable visual quality** against previous methods for 10 minutes and an hour of optimization, which raises concerns about the misalignment between the visual quality and the quantitative metrics.

**Rendering Efficiency**: We provide the rendering speed of our method against previous methods. Our method reaches real-time inference speed (100+ FPS for rendering 512×512 images), which is hundreds of times faster than the previous ones. Our method takes considerably less memory than the previous ones.

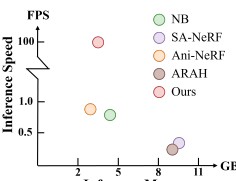

## 4.3 APPLICATIONS

After training, we can export mesh representations, which enables instant downstream applications. We showcase two examples of novel pose synthesis, material editing, and human relighting in Figure 1. For more examples, please refer to our **supplementary**, especially **the accompanied video**.

## 5 CONCLUSIONS

We present EMA, which learns human avatars through hybrid meshy neural fields efficiently. EMA jointly learns hybrid canonical geometry, materials, lights, and motions via a rasterization-based differentiable renderer. It only requires one hour of training and can render in real-time with a triangle renderer. Minutes of training can produce plausible results. Our method enjoys flexibility from implicit representations and efficiency from explicit meshes. Experiments on the standard benchmark indicate the competitive performance and generalization results of our method. The digitized avatars can be directly used in downstream tasks. We showcase examples including novel pose synthesis, material editing, and human relighting.

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
