# OpenReview forum: "Efficient Meshy Neural Fields for Animatable Human Avatars"
_ICLR.cc/2024/Conference — Submitted to ICLR 2024_

### Official Review · Reviewer_QZey · 2023-10-20

**Soundness:** 2 fair
**Presentation:** 2 fair
**Contribution:** 2 fair
**Rating:** 6
**Confidence:** 4

**Summary:**

This paper presents a novel method for reconstructing animatable human avatars from videos. The proposed method, named EMA, models canonical shapes, materials, lights and motions separately using different neural fields. With an analysis by synthesis framework, those terms can be optimized using image level losses via a differentiable marching tetrahedra algorithm. A mesh-based representation can be distilled from this representation, which greatly improves its rendering efficiency. Extensive comparisons are conducted with previous methods. Improved results are observed on the H36M dataset and comparable results are observed on ZJUMOCAP.

**Strengths:**

* Extensive experimental results.

This paper conducted many experiments with comparisons with previous baselines and detailed analysis.

**Weaknesses:**

* Better illustrations are needed.

Many illustrative figures in this manuscript lack proper notation and are hard to align with the text. It would make the reader understand better about the technical details better if more details were included.

**Questions:**

* Novelty?

The proposed method combines existing methods [1] and [2] (and many other papers in the field). Although one of the major differences is the efficiency in rendering, the same improvement can also be theoretically achieved by training a SDF and extracting a water-tight mesh for rendering afterward. It would be great to have a more thorough discussion on the merit of the current pipeline as well as other potential advantages it brings.









* Comparison with other efficient frameworks?

On the efficiency side, the joint optimization of mesh and SDF is interesting. However, there are also many other sparse structures used for speeding up the rendering efficiency like using layered mesh representation [] and sparse volumes [3, 4]. It would enhance the quality of the manuscript if some comparisons or discussions were added.


[1] Jacob Munkberg, Wenzheng Chen, Jon Hasselgren, Alex Evans, Tianchang Shen, Thomas Mu ̈ller, Jun Gao, and Sanja Fidler. Extracting triangular 3d models, materials, and lighting from images. In CVPR, pp. 8270–8280, 2022.
[2] Chung-Yi Weng, Brian Curless, Pratul P. Srinivasan, Jonathan T. Barron, and Ira Kemelmacher- Shlizerman. Humannerf: Free-viewpoint rendering of moving people from monocular video. In CVPR, pp. 16189–16199, 2022.
[3] Tianjian Jiang, Xu Chen, Jie Song, and Otmar Hilliges. "Instantavatar: Learning avatars from monocular video in 60 seconds." In Proceedings of the IEEE/CVF Conference on Computer Vision and Pattern Recognition, pp. 16922-16932. 2023.
[4] Edoardo Remelli, Timur Bagautdinov, Shunsuke Saito, Chenglei Wu, Tomas Simon, Shih-En Wei, Kaiwen Guo et al. "Drivable volumetric avatars using texel-aligned features." In ACM SIGGRAPH 2022 Conference Proceedings, pp. 1-9. 2022.
[5] Donglai Xiang, Fabian Prada, Timur Bagautdinov, Weipeng Xu, Yuan Dong, He Wen, Jessica Hodgins, and Chenglei Wu. "Modeling clothing as a separate layer for an animatable human avatar." ACM Transactions on Graphics (TOG) 40, no. 6 (2021): 1-15.

---

> ### Author Response · Authors · 2023-11-22
> **Response to the Reviewer QZey**
>
> We thank the reviewer for the valuable and constructive feedback.
>
>
> **[Q1]** Illustration of the model.
>
> We have revised the figure accordingly, including annotations to help clarify our design. The revised version has been updated.
>
> **[Q2]** Novelty.
>
> [1] is designed for static object reconstruction. Our method extends it to modeling dynamic humans. One insight is that, given the mesh extraction framework from [1], we can directly learn the forward LBS skinning without iterative and (potentially) ambiguous root-finding [3].
> While [2] uses learned backward skinning which is insufficient for animation and faithful reconstruction of the motion (We include a figure in the updated supp. to demonstrate the problems).
> To the best of our knowledge, this work is the first to explore the pipeline of differentiable textured mesh + motions (forward LBS skinning + non-rigids) and demonstrate its merits like fast training, quick inference, mesh representation, and template-free property.
>
> **[Q3]** Extracting mesh for NeRF-based methods.
>
> Every time a new pose is given, the NeRF-based methods have to predict new density and color fields for mesh extraction, which is inefficient. One can only extract the canonical mesh, yet the skinning weights are either inaccessible for backward-skinning ones or maybe (partially) wrong after discretization, and the non-rigid motions may be lost or degraded due to discretization. Besides, the threshold and resolution of the marching cube algorithm are tricky to tweak for both good shape and appearance quality. To summarize, the post-processing of NeRFs leads to either quality degradation or loss of features (like non-rigid).
>
> Additionally, we can deem our method as a mesh-, or quantization-aware framework for modeling non-rigid. Moreover, the learned motion dynamics not only help inference rendering but also facilitate the training as a result of better inter-frame correspondences, leading to better quality.
>
> **[Q4]** More comparisons.
>
> InstantAvatar [3]  is designed for learning a neural field from a fixed-posed human-rotating monocular video or a synthetic monocular video without loose clothes, and it is unknown that it can handle non-rigid dynamics with large poses from multi-view datasets. We also do not find any qualitative results from InstantAvatar for extensive comparison. Furthermore, our rendering speed (100 FPS) is faster than InstantAvatar (15 FPS).
>
> As for [4] and [5], they are very exciting industrial-standard research works that push the frontier of human digitization. However, the evaluation setting of [4] is unclear and [4] unfortunately does not conduct extensive comparisons of the benchmarks. The code of [5] is unavailable.
>
> Ultimately, we have referred to the related literature and uploaded the newer version of the pdf.
>
>
> [1] Jacob Munkberg, Wenzheng Chen, Jon Hasselgren, Alex Evans, Tianchang Shen, Thomas Mu ̈ller, Jun Gao, and Sanja Fidler. Extracting triangular 3d models, materials, and lighting from images. In CVPR, pp. 8270–8280, 2022.
>
> [2] Chung-Yi Weng, Brian Curless, Pratul P. Srinivasan, Jonathan T. Barron, and Ira Kemelmacher- Shlizerman. Humannerf: Free-viewpoint rendering of moving people from monocular video. In CVPR, pp. 16189–16199, 2022.
>
> [3] Tianjian Jiang, Xu Chen, Jie Song, and Otmar Hilliges. "Instantavatar: Learning avatars from monocular video in 60 seconds." In Proceedings of the IEEE/CVF Conference on Computer Vision and Pattern Recognition, pp. 16922-16932. 2023.
>
> [4] Edoardo Remelli, Timur Bagautdinov, Shunsuke Saito, Chenglei Wu, Tomas Simon, Shih-En Wei, Kaiwen Guo et al. "Drivable volumetric avatars using texel-aligned features." In ACM SIGGRAPH 2022 Conference Proceedings, pp. 1-9. 2022.
>
> [5] Donglai Xiang, Fabian Prada, Timur Bagautdinov, Weipeng Xu, Yuan Dong, He Wen, Jessica Hodgins, and Chenglei Wu. "Modeling clothing as a separate layer for an animatable human avatar." ACM Transactions on Graphics (TOG) 40, no. 6 (2021): 1-15.

---

> ### Author Response · Authors · 2023-11-23
> **Looking forward to your feedback**
>
> Dear Reviewer QZey,
>
> Thanks again for your valuable advice and supportive comments! We have responded to your initial comments. We are looking forward to your feedback and will be happy to answer any further questions you may have.

---

### Official Review · Reviewer_imbQ · 2023-10-30

**Soundness:** 3 good
**Presentation:** 3 good
**Contribution:** 3 good
**Rating:** 6
**Confidence:** 4

**Summary:**

The paper proposes a method for learning articulable human avatar models from image data. This is accomplished by modeling the representation as a signed distance function and jointly optimizing the canonical shape, material and lights (for appearance), and skinning weights (for motion). Because the representation is modeled as a signed distance function, it can be extracted into a textured mesh and animated with learned skinning weights in order to efficiently render the human in new poses. The result quality is compared to a number of baseline works, including those optimizing textured meshes directly, and those training volumetric representations. It is demonstrated that the proposed method leads to better quality, along with the ability to change lighting and train significantly more efficiently.

**Strengths:**

In my opinion, the strengths of the method are as follows:
1. The paper is described clearly, and the method makes intuitive sense. Optimizing the representation's geometry (SDF) and appearance (materials and lighting), and motion (skinning weights) jointly seems like the correct approach to learn an avatar from only image data. The optimization objective proposed makes sense and seems well-posed for solving this under-constrained task.
2. The proposed retains additional control over various factors which is unlike the existing methods. For example, volumetric methods do not model materials and lighting and thus cannot support relighting. This allows the proposed method to be applied in different applications which are not possible for other methods.

**Weaknesses:**

In my opinion, the weaknesses of the method are as follows:
1. I view the comparisons as not being extensive. For example, this concurrent work seems to solve the same problem [1], and they compare to other baselines which appear better than the ones used in this method. Additionally, there exist a number of methods which use the SMPL mesh for skinning weights driving motion [2][3] for which the quality appears quite good. Why are these approaches not compared to in this work, as it appears that these methods are able to generate better results qualitatively? If this method is focusing on human body avatars, why learn general skinning weights with a skeleton instead of using those from a known human body model, such as SMPL.
2. For the efficient training, I'm not sure why the representation actually trains faster? It seems to use the entire image as opposed to individual rays as in volume rendering, and marching to find the surface also requires a number of samples of the SDF representation. This results in each iteration having more rays, and just as many samples per ray, so I don't understand why the method actually trains faster.

[1] https://lukas.uzolas.com/Articulated-Point-NeRF/

[2] https://machinelearning.apple.com/research/neural-human-radiance-field

[3] https://tijiang13.github.io/InstantAvatar/

**Questions:**

I have no additional questions on the manuscript. Overall, the paper proposes a method which makes sense and learns human avatars which have capabilities that other volumetric methods are not capable of, such as relighting, efficient rendering and training, and compatibility with graphics pipeline. However, I do not understand why the existing state-of-the-art methods in generating avatars have not been compared to. If it is because they use the SMPL template for driving the motion of the human avatars, I don't view this to be a limitation, since this paper also focuses on humans. Understanding why these methods were not included in the comparisons, or adding them as comparisons, would significantly strengthen the paper and lead me to increase my score.

**Update after author response**

After reading the author response, I still remain borderline on the paper. I understand not comparing to methods which are designed for reconstructing from a monocular video, and appreciate the addition of some comparisons here. Additionally, I appreciate the additional timing results. I have thus increased my score a bit for the paper.

However, the justification for not using the SMPL model does not seem convincing to me. If the reason is non-rigid deformations such as clothing, then it needs to be explicitly demonstrated in the paper that this is improved by the proposed method. Additionally, other contributions (such as the correction MLP from Neuman) give the ability to model these types of clothing deformations with the SMPL weights.

---

> ### Author Response · Authors · 2023-11-22
> **Response to the Reviewer imbQ**
>
> We thank the reviewer for the constructive and informative review.
>
>
> **[Q1]** More Comparison.
>
> Our method focuses on human body avatars, which is different from the aim of Articulated Point NeRF [1]. So we do not compare the general dynamic NeRF baselines in Articulated Point NeRF.
>
> Both Neuman [2] and InstantAvatar [3] focus on monocular video reconstruction, and do not compare themself to the standard multi-view video benchmarks like H36M and ZJU-Mocap. So directly comparing with their methods is unfair.
>
> Neuman [2] is designed for learning human NeRF and scene NeRF from a single-view video, and it only learns a frame-dependent error-correction network for observation space. So quantitative comparison is unfair. We give a qualitative comparison of ZJU-Mocap in the updated supplementary. Our results preserve more details than Neuman.
>
> InstantAvatar [3]  is designed for learning a neural field from a fixed-posed human-rotating video or a synthetic video without loose clothes, and it is unknown that it can handle non-rigid dynamics with large poses from multi-view datasets. We also do not find any qualitative results from InstantAvatar for extensive comparison. Furthermore, our rendering speed (100 FPS) is faster than InstantAvatar (15 FPS).
>
>
> Ultimately, we have referred to the related literature and uploaded the newer version of the pdf.
>
>
>
> **[Q2]** Using existing skinning templates or not.
>
> It depends. We learn geometry, appearance, and motions (skinning and non-rigid) to provide a **template-free** solution to avatar reconstruction. The problem with using templates is that:
>
> 1. The templates are usually proprietary in real-world usage.
> 2. There exists a gap between the skinning field of clothed humans and naked humans. Our skinning field can fill the gap via optimization.
> 3. Templates are object-specific (e.g., mature humans), which hinders their wider application to other kinds of objects.
>
> However, there are other factors to be considered. For example, if there are limited variety of motions in the training data, the quality of the skinning field may be poor due lack of inter-frame regularization. Besides, Leveraging templates is a good choice to initialize or regularize the skinning field and speed up convergence [2,4].
>
>
>
>
> **[Q3]** Efficiency breakdown.
>
>
> The efficiency of the method comes from two-fold: Both geometry and rendering. We provide a runtime breakdown and analyze the efficiency.
> The runtime breakdown:
>
> | Function                 | Time (ms)               |
> |---------------------|---------------------|
> | Extract Mesh (w/ geo. NF query, w/ non-rigid NF query)      | 11.39273 |
> | Extract Mesh (w/o geo. NF query, w/ non-rigid NF query)       | 3.08209  |
> | Render Mesh (w/ texture NF query, w/ env light query)   | 7.00726  |
> | └──  (texture NF query)      | 4.08976  |
>
> - "NF" means Neural Fields.
> - Extract Mesh: For NF query in mesh extraction, we query both the canonical SDF field and the non-rigid motion field.
>     -  In inference time, we only query the neural field once to get the canonical mesh and use the same mesh for the latter rendering. The motion field will be queried by the number of vertex of the mesh times.
>     -  In training time, we need to query the NF for every optimization step to update the SDF and non-rigid fields.
>     -  The insight is that, compared with NeRFs, our gradients only flow through the iso-surface, which is drastically less than NeRF, which where the gradients flow over the whole space.
> - Render Mesh:
>     - For rendering an NxN resolution image, we only query **O(NxN)** times for the texture field thanks to the rasterization. While NeRF-based method needs to query **O(NxNxM)**, where M is the number of points per pixel (ray). The usage of tiny-cuda-nn, a highly optimized package for neural fields, further improves the speeds.
>     - It involves both texture querying and env-light map querying.
>
>
>
> [1] https://lukas.uzolas.com/Articulated-Point-NeRF/ (https://lukas.uzolas.com/Articulated-Point-NeRF/)
>
> [2] https://machinelearning.apple.com/research/neural-human-radiance-field (https://machinelearning.apple.com/research/neural-human-radiance-field)
>
> [3] https://tijiang13.github.io/InstantAvatar/ (https://tijiang13.github.io/InstantAvatar/)
>
> [4] https://github.com/taconite/arah-release(https://github.com/taconite/arah-release)

---

> ### Author Response · Authors · 2023-11-23
> **Looking forward to your feedback**
>
> Dear Reviewer imbQ,
>
> Thanks again for your valuable advice and supportive comments! We have responded to your initial comments. Please feel free to let us know if you have any further questions.

---

### Official Review · Reviewer_PTA7 · 2023-10-31

**Soundness:** 4 excellent
**Presentation:** 3 good
**Contribution:** 3 good
**Rating:** 6
**Confidence:** 3

**Summary:**

This paper introduces a new framework for modeling dynamic human avatars with neural fileds. In this work, three different group of neural fields are adopted to record shape, material and motion information respectively and optimized jointly. Specifically, this work employs an 8-layer MLP to model the SDF of the canonical shapes. A 2-layer MLP with hash-encoding is adopted for efficient material query. SNARF is introduced for skinning weights, and another 4-layer MLP is used for non-rigid modeling. Finally, these neural fields are integrated under the Linear Skining framework and further rendered by a differentiable renderer to fit target images. Experimental results demonstrate that this new framework can achieve superior rendering quality with less training and inference times.

**Strengths:**

+ The proposed method is technically sound. It is clever to integrating LBS model and PBR materials within Neural Fields for high-quality rendering.

+ Employing environment lights and non-rigid models is also a good way to enhance the rendering results.

+ The experimental results are convincing. This work achieves better results with less training and inference time.

**Weaknesses:**

- It is difficult to read the main maniscript without supplementary materials. For example, crucial information such as Nerf structure should be included in the main maniscript.

- This work would be further strengthened if more in-the-wild results were provided.

**Questions:**

Here are some concerns:

1. As the framework takes into account environmental lights and the non-rigid model, how does this method perform on in-the-wild data?

2. Can this method be used for avatar modeling with soft cloth (e.g. wearing a dress)?

3. What is the geometric precision of this method?

---

> ### Author Response · Authors · 2023-11-22
> **Response to the Reviewer PTA7**
>
> We thank the reviewer for the positive and constructive feedback.
>
>
> **[Q1]** Illustration of the model.
>
> We have revised the figure accordingly, including annotations to help clarify our design. The revised version has been updated.
>
>
> **[Q2]** In-the-wild data.
>
> The challenge of the in-the-wild data are 1) inaccurate pose tracking and parsing, and 2) inaccurate global coordinate system.
>
> We use datasets that are in the controlled environment, as this paper focuses on building a pipeline that enjoys fast training, quick inference, mesh representation, template-free (which can be extended for a variety of objects and avoid proprietary models), and feedforward LBS for training.
>
> Recently, [1] proposed an in-the-wild dataset. We will explore the in-the-wild scenario in the future.
>
>
> **[Q3]** Soft clothes.
>
> Yes, our method can be applied to soft clothes thanks to the non-rigid modeling and the template-free property. In our supp. video, there are avatars with loose clothes in clip 01:06 - 01:24 (or "ema.supp.representation_visualization.mp4" in supp). The mesh visualization offers a better view. Admittedly, modeling clothes dynamics like dress is a hard research problem requiring further research.
>
>
> **[Q4]** Geometry quality.
>
> In supplementary materials, Sec. H, Mesh visualization, we qualitatively visualize the canonical meshes. Note that the number of faces for each mesh is quite small. Though increasing the resolution of tetrahedra grids may improve the details of both geometry and materials, we do not conduct this experiment for it is orthogonal to our technical contributions.
>
>
> [1] Kaufmann, M., Song, J., Guo, C., Shen, K., Jiang, T., Tang, C., Zárate, J.J. and Hilliges, O., 2023. EMDB: The Electromagnetic Database of Global 3D Human Pose and Shape in the Wild. In Proceedings of the IEEE/CVF International Conference on Computer Vision (pp. 14632-14643).

---

> ### Author Response · Authors · 2023-11-23
> **Looking forward to your feedback**
>
> Dear Reviewer PTA7,
>
> Thanks again for your valuable advice and supportive comments! We have responded to your initial comments. We are looking forward to your feedback and will be happy to answer any further questions you may have.

---

### Official Review · Reviewer_XfaE · 2023-11-01

**Soundness:** 2 fair
**Presentation:** 2 fair
**Contribution:** 2 fair
**Rating:** 5
**Confidence:** 4

**Summary:**

This paper proposes a method called EMA (Efficient Meshy neural fields for Animatable human Avatars) for efficiently generating animatable human avatars from videos. The main goal is to overcome the shortcomings of existing volume rendering-based methods in terms of training and inference speed and to achieve compatibility with rasterization renderers for direct application to downstream tasks.The EMA method jointly optimizes explicit triangular canonical mesh, spatially varying materials, and motion dynamics through end-to-end inverse rendering. These components are encoded by separate neural fields, eliminating the need for preset human templates, rigging, or UV coordinates. The authors also use differentiable rasterization techniques to learn mesh properties and forward skinning, improving the efficiency of the method.Compared to existing methods, the EMA method has significant advantages in training and inference speed. It is highly compatible with rasterization renderers, has a short training time, and fast rendering speed. Experimental results show that the EMA method has competitive performance in novel view synthesis, generalization to novel poses, and training time and inference speed.

**Strengths:**

The EMA method achieves real-time rendering through efficient mesh rendering. Moreover, it computes the loss on the entire image, and gradients flow only on the mesh surface, resulting in improved training speed. Compared to volume rendering methods, the EMA method has a shorter training time.

**Weaknesses:**

1.	The EMA method employs neural networks to encode canonical geometry, materials, and motion models. In practical applications, the complexity of these neural networks may affect the inference speed. Have the authors conducted experiments in this regard, or attempted to use simpler or more efficient neural network architectures to balance the inference speed and reconstruction quality?
2.	The EMA method learns a fixed environment light and uses a Physically-Based Rendering (PBR) material model. In practical applications, would this lighting and material modeling approach potentially limit the inference speed to some extent? Is it possible to use simpler lighting and material models to further improve efficiency?
3.	The EMA method employs pose-dependent non-rigid offsets to compensate for non-rigid cloth dynamics. Does this modeling approach increase computational complexity and impact inference speed? Are there any other more efficient methods to handle non-rigid cloth dynamics? Judging from the video results, it seems that EMA might also be limited in terms of non-rigid cloth dynamic modeling?
4.	The EMA method relies on skeletal pose tracking of the input video. So is it conceivable that the accuracy of pose tracking might affect inference speed and result quality? Has the author considered the performance and speed of the method in the case of inaccurate pose tracking, I think this will be of great help in practical applications?
5.	In the experimental part, it can be seen that in the case of fast training (10 minutes), EMA has a satisfactory result (better than ARAH). However, under the same hour, the advantage seems not obvious. And as I said before, when the pose tracking quality is poor, the EMA is also greatly affected.
6.	In addition, are there other methods that also use mixed representation training methods? The improvements brought by this method have not been well explained and proven in experiments.

**Questions:**

As stated above.

**Comments after Rebuttal**

I thank the authors for the reply. However, several issues still exist, which makes this paper a borderline one to me. For example, the quantitative comparisons with the baseline of ARAH are not that strong to me. Although the authors show some visual comparisons, the insufficient comparison samples make the improvement of this work weaker. Besides, as also noted by other reviewer colleagues, the justification of in-the-wild results and not using SMPL seems unclear to me. I thus keep my original rating.

---

> ### Author Response · Authors · 2023-11-22
> **Response to the Reviewer XfaE (Part 1)**
>
> We thank the reviewer for the detailed and constructive reviews.
>
> **[Q1,Q2,Q3]** The efficiency of each module.
>
> We provide a runtime breakdown and analyze the efficiency.
>
> The runtime breakdown:
>
> | Function                 | Time (ms)               |
> |---------------------|---------------------|
> | Extract Mesh (w/ geo. NF query, w/ non-rigid NF query)      | 11.39273 |
> | Extract Mesh (w/o geo. NF query, w/ non-rigid NF query)       | 3.08209  |
> | Render Mesh (w/ texture NF query, w/ env light query)   | 7.00726  |
> | └──  (texture NF query)      | 4.08976  |
>
> - "NF" means Neural Fields.
> - Extract Mesh: For NF query in mesh extraction, we query both the canonical SDF field and the non-rigid motion field.
>     -  In inference time, we only query the neural field once to get the canonical mesh and use the same mesh for the latter rendering. The motion field will be queried by the number of vertex of the mesh times.
>     -  In training time, we need to query the NF for every optimization step to update the SDF and non-rigid fields.
> - Render Mesh:
>     - For rendering an NxN resolution image, we only query O(NxN) times for the texture field thanks to the rasterization. While NeRF-based methods need to query O(NxNxM), where M is the number of points per pixel (ray). The usage of tiny-cuda-nn [4], a highly optimized package for neural fields, further improves the speeds.
>     - It involves both texture querying and env-light map querying.
>
>
> **[Q1]** Efficiency wrt. NNs for geometry, appearance, and motions as they may affect inference speed.
>
> We follow the design of NN of the previous arts. Since they are fast in our pipeline, we do not further tweak or improve the architecture of the fields. Besides, during inference, the mesh is only extracted once. So there is no NF overhead for mesh extraction. For light and appearance, we only query O(NxN) thanks to the rasterization technique.
>
> **[Q2]** Efficiency wrt. lighting and PBR.
>
> It is possible to use simpler ones like Spherical harmonic lighting [1], but it is not necessary as the computation overhead is, again, negligible. In addition, PBR and Env light are industrial standards in Computer Graphics, e.g., they are used in video games to achieve real-time and photo-realistic rendering [2].
>
>
> **[Q3]** Efficiency wrt. modeling dynamics
>
> We model dynamics for better reconstruction and animation. Since we only move the coordinates of the extracted mesh, the time is negligible. To the best of our knowledge, there is no literature providing even efficient approach to modeling the clothes dynamics. In our supp. video, there are non-rigid offsets in clip 01:06 - 01:24 (or "ema.supp.representation_visualization.mp4" in supp). The mesh visualization offers a better view. Admittedly, modeling clothes dynamics is a hard research problem requiring further research.
>
>
> **[Q4]** The affection of inaccurate pose tracking.
>
> We show the results on the synthetic data with poses from ZJU-MoCap.
>
> | noise | novel view |  | novel pose |  |
> | --- | --- | --- | --- | --- |
> | scale | psnr | ssim | psnr | ssim |
> | 0.05 | 24.41 | 0.931 | 23.43 | 0.913 |
> | 0.02 | 24.42 | 0.931 | 23.48 | 0.913 |
> | 0.01 | 24.43 | 0.932 | 23.48 | 0.914 |
> | 0.005 | 24.35 | 0.931 | 23.68 | 0.917 |
> | 0.0 | **25.98** | **0.950** | **25.15** | **0.938** |
>
> For quality side:
> - Adding noises to the training poses leads to performance degradation.
> For speed side:
> - For training, they spent almost the same convergence time.
> - For inference, the noisy poses do not affect the speed.
>
>
> The paper focuses on building a pipeline that enjoys fast training, quick inference, mesh representation, template-free (which can be extended for a variety of objects and avoid proprietary models), and feedforward LBS for training. Therefore, we use datasets that are in a controlled environment.
>
> How to overcome inaccurate pose tracking is an active research direction.
> Recently, [3] proposed an in-the-wild dataset. We will explore the in-the-wild scenario in the future.

---

> ### Author Response · Authors · 2023-11-22
> **Response to the Reviewer XfaE (Part 2)**
>
> **[Q5]** Compare with ARAH.
>
> Although our method only achieves comparative performance with ARAH, the qualitative results of our method are better in Fig. 1 and Fig. 2, e.g., the rendering of ARAH is blurrier than ours. This indicates the drawback of current reference-based metrics.
>
>
> **[Q6]** Other mixed representation.
>
> We refer to and discuss other mixed representation methods in the related works. The majority of them are volume-based, while our method is mesh-based. Compared with previous methods, the training speed is largely increased, the rendering is real-time, and the outputs are triangular meshes that are fully compatible with the industrial graphics pipeline. Here, the insight is that equipped with a differentiable meshes extractor and renderer, not only the geometry and appearance can be learned, but the motions, both rigid and non-rigid ones, can be learned in a feed-forward manner, which avoids proprietary human templates.
>
>
>
> [1] Ramamoorthi, R. and Hanrahan, P., 2001, August. An efficient representation for irradiance environment maps. In Proceedings of the 28th annual conference on Computer graphics and interactive techniques (pp. 497-500).
>
> [2] Advances in Real-Time Rendering in Games, 2022, https://advances.realtimerendering.com/s2022/index.html
>
> [3] Kaufmann, M., Song, J., Guo, C., Shen, K., Jiang, T., Tang, C., Zárate, J.J. and Hilliges, O., 2023. EMDB: The Electromagnetic Database of Global 3D Human Pose and Shape in the Wild. In Proceedings of the IEEE/CVF International Conference on Computer Vision (pp. 14632-14643).
>
> [4] Muller, T., tiny-cuda-nn, https://github.com/NVlabs/tiny-cuda-nn

---

> ### Author Response · Authors · 2023-11-23
> **Looking forward to your feedback**
>
> Dear Reviewer XfaE,
>
> Thanks again for your valuable advice and supportive comments! We have responded to your initial comments. Please feel free to let us know if you have any further questions.

---

### Author Response · Authors · 2023-11-22
**Response to the Reviewers**

Dear Reviewers,

We would like to express our gratitude for your valuable time and effort spent in providing detailed and constructive feedback on our paper. We have taken all your comments into consideration and have made necessary revisions to our paper as well as supplementary materials.

Please kindly note that we have responded to each of your queries and concerns separately. We have also updated our paper and supplementary materials in line with the recommendations given by you in the comments. We hope these updates have improved the clarity and completeness of our work.

We kindly invite you to review our responses and the revised version of our work. We expect that these updates will address your concerns and improve the quality of our paper.

Again, we appreciate your invaluable feedback and look forward to your further comments and suggestions.

Best regards,

The Authors of 7093

---

### Meta-Review · Area_Chair_ovZY · 2023-12-10

**Metareview:**

(a) The paper presents EMA for digitizing animatable human avatars from videos. It optimizes separate neural fields for triangular canonical mesh, materials, and motion dynamics, enabling efficient training and real-time rendering. EMA addresses limitations of volume rendering-based methods by reducing optimization times and improving inference speed. Its compatibility with rasterization renderers and the disentanglement of meshes allow for versatile downstream applications, including pose and material editing, and relighting. Extensive experiments demonstrate EMA's competitive performance and speed advantage.

(b) Strengths:
1. Innovative optimization of separate neural fields.
2. Good improvements in training and rendering efficiency.
3. High compatibility with rasterization renderers.
4. Disentanglement feature facilitating downstream applications.

(c) Weaknesses:
1. The Reviewers, especially XfaE, felt that the quantitative comparisons with baseline methods like ARAH weren't convincing enough. Comparison to stronger baselines such as Neuman and GP-Nerf is required.

2. Both Reviewers PTA7 and XfaE highlighted the absence of in-the-wild testing results, which are crucial for demonstrating the robustness and practical applicability of EMA in less controlled, real-world scenarios.

3. Reviewer imbQ was not fully convinced by the authors' justification for not using established models like SMPL, particularly in the context of modeling non-rigid deformations such as clothing.

**Justification For Why Not Higher Score:**

1. The Reviewers, especially XfaE, felt that the quantitative comparisons with baseline methods like ARAH weren't convincing enough. Comparison to stronger baselines such as Neuman and GP-Nerf is required.

2. Both Reviewers PTA7 and XfaE highlighted the absence of in-the-wild testing results, which are crucial for demonstrating the robustness and practical applicability of EMA in less controlled, real-world scenarios.

3. Reviewer imbQ was not fully convinced by the authors' justification for not using established models like SMPL, particularly in the context of modeling non-rigid deformations such as clothing.

**Justification For Why Not Lower Score:**

N/A

---

### Decision · Program_Chairs · 2024-01-16

Reject